# The Relationship between Executive Functions and Gross Motor Skills in Rural Children Aged 8–10 Years

**DOI:** 10.3390/healthcare10040616

**Published:** 2022-03-25

**Authors:** Zahra Fathirezaie, Sérgio Matos, Elham Khodadadeh, Filipe Manuel Clemente, Georgian Badicu, Ana Filipa Silva, Seyed Hojjat Zamani Sani, Samaneh Nahravani

**Affiliations:** 1Physical Education and Sport Science Faculty, University of Tabriz, Tabriz 51666, Iran; khodadadehelham1995@gmail.com (E.K.); hojjatzamani8@gmail.com (S.H.Z.S.); samanehnahravani@gmail.com (S.N.); 2Escola Superior Desporto e Lazer, Instituto Politecnico de Viana do castelo, Rua Escola Industrial e Comercial de Nun Alvares, 4900-347 Viana do Castelo, Portugal; sergioms@esdl.ipvc.pt (S.M.); anafilsilva@gmail.com (A.F.S.); 3Research Center in Sports Performance, Recreation, Innovation and Technology (SPRINT), 4960-320 Melgaço, Portugal; filipe.clemente5@gmail.com; 4Department of Sports, Higher Institute of Educational Sciences of the Douro, 4560-708 Penafiel, Portugal; 5Instituto de Telecomunicacoes, Delegacao da Covilha, 1049-001 Lisboa, Portugal; 6Department of Physical Education and Special Motricity, Faculty of Physical Education and Mountain Sports, Transilvania University of Braşov, 500068 Braşov, Romania; georgian.badicu@unitbv.ro; 7The Research Centre in Sports Sciences, Health Sciences and Human Development (CIDESD), 5001-801 Vila Real, Portugal

**Keywords:** executive functions, gross motor skills, rural children

## Abstract

Considering that cognitive and motor dimensions of human beings grow together, and that primary school age is one of the most important stages of children’s cognitive and motor development, the aim of this study was to investigate the relationship between executive functions and gross motor skills in rural children aged 8–10 years. This descriptive and correlational research was conducted with 93 Iranian rural primary school children aged 8 to 10 years. A Behavior Rating Inventory of Executive Function (BRIEF) questionnaire and the Test of Gross Motor Development, second edition (TGMD-2) were used to collect data on executive functions and gross motor skills, respectively. The results showed that most of the correlations between criterion and predictor variables were moderate. In the regression results we observed that among the components of executive functions, inhibition, working memory, planning/organizing, and organization had a significant relationship with gross motor skills, but no relationship was found between other components and motor skills. As a result, it can be said that in predicting cognitive development and specifically mentioned executive functions, gross motor skills are an important and effective factor among rural children and, given the importance of cognitive development and executive functions in childhood, it seems that by helping to develop their gross motor skills, executive functions will also be strengthened. Finally, possible future studies are addressed, which could investigate the effect of different aspects of motor skill classifications on executive functions.

## 1. Introduction

Successful functioning throughout life (e.g., in school, social interactions, and physical and mental health) requires well-developed executive functions (EFs) [1]. The term “EFs” refers to a multifaceted concept that has been defined in a variety of ways. Nonetheless, there are some common components in these definitions that highlight the important qualities of EFs. In summary, EFs are cognitive processes that influence actions, ideas, and emotions from the top down [2]. Second, EFs are only activated in instances where conscious, goal-directed conduct is required, not in situations where automatic or intuitive conduct is required, so employing EFs necessitates attempt and effort [2,3]. Third, EFs refer to a collection of related but distinct cognitive processes, making it a multidimensional, rather than unitary, concept [1]. Based on recent studies, EFs are related to different brain regions [4]. Takeuchi et al. [5] used the terms hot and cool executive functions, stating that hot executive functions include affective decision making or decision making about events that have emotionally significant consequences and are associated with the orbital regions of the prefrontal cortex; cool executive functions include working memory, planning, and problem solving and are related to the dorsolateral regions of the prefrontal cortex [5]. This viewpoint is supported by an increasing corpus of neurophysiological and neuroimaging evidence [6]. Neuroimaging techniques have revealed that some motor and cognitive regions critical for motor performance and cognition, such as the cerebellum, dorsolateral prefrontal cortex, and linking structures (including the basal ganglia), are co-activated, so evidence of the relationship between motor and cognitive development can also be found in behavioral studies [7]. In this regard, Piaget believed that motor and cognitive skills are intertwined. Piaget’s hypothesis was based on the premise that children learn from observable object motor activities [8,9]. Payne and Isaacs [10] stated that cognitive and motor development interact with each other and inhibit or help each other throughout life. However, research on the relationship between cognitive and motor development is limited to specific cognitive domains [11], such as working memory [12], language [13], etc.

Development of gross motor skills as part of motor development is important in young children to aid in developmental functions such as perceptual and cognitive abilities [14]. Gross motor skills are the first motor skills to be developed, and they are important for physical health, mental cognition, and social adaption [15]. Different perspectives on the relationship between motor skills and cognitive skills in children have existed in the past. On the one hand, motor and cognitive skills have long been thought of as distinct processes that develop separately and involve different brain regions [9,16]. In this regard, and rejecting the relationship with the study of different dimension effects of human development, especially cognitive and motor development, several studies have been conducted [12,17]. For example, research on the relationship between cognitive development and gross motor skills [17], as well as executive functions and motor skills [12], showed that there is a significant relationship between cognition and movement. Diamond [18] found that the connection between motion and cognition was more than she had imagined [18]. With the help of neural imaging techniques, she provided strong evidence for a link between fine motor skills and cognitive skills and it was shown that the forearm, cerebellum, and meninges are activated during cognitive tasks, as well as motor skills.

Moreover, the community in which the research was performed is remarkable. Some studies have confirmed that parents’ socioeconomic status and the child’s living environment may be related to children’s motor development [19,20]. Therefore, rural communities of children should be considered, especially in developing countries. Due to their environmental nature, children from rural areas may have different opportunities to develop their skills [19]. Considering the living environment, rural children have a wider space to play than urban children do. Although rural children usually live in normal houses, not apartments, they have less access to a park’s playground equipment than urban children, and spend most of their time free playing in open spaces with natural tools, such as wood, soil, etc. It has been shown that EFs may be developed by playing in natural environments [21].

Given that executive functions [22,23] and gross motor skills [24,25] are related to both the health and education of children, and that to date there has been no known comprehensive review specifically examining the relation between physical activity and cognitive outcomes in rural children, as well as the fact that children’s cognitive and motor dimensions grow together [26], this study considered the relationship between executive functions and gross motor skills in rural children aged 8–10 years.

## 2. Materials and Methods

### 2.1. Subjects and Design

The present study was a post-event correlational study that was conducted in its field of application. The statistical population of this study was healthy primary school children aged 8 to 10 years (mean = 9.10 ± 0.767) in rural Iran. Two primary schools for girls and boys were selected by a cluster random sampling method and 93 students aged 8 to 10 years (59 girls and 34 boys) participated in the study. The number of samples was estimated based on G*Power 3.1.9.4 software. Two predictor variables were independently used as criterion variables, which included object control and locomotor. Additionally, the effect size was estimated to be 0.15 based on the literature review, the alpha error probability was 0.05, and the power of the test was 0.95. Therefore, the total sample size was 74 in the one-tailed test and 89 in the two-tailed test; in this study, 93 subjects participated.

Firstly, a consent letter was received from parents for their children’s participation in the research, and all participants voluntarily took part within the present study. In addition, in order to assess students for cognitive, behavioral, and motor problems, we first obtained the consent of the students’ parents and the relevant school principal, and then used the self-report forms available in the school. Finally, students who had no behavioral or cognitive problems participated in this research.

### 2.2. Procedure

First, the necessary coordination was completed with the education and school management departments. After obtaining permission, the research process began in one of the sport clubs, which was an accessible and suitable space for the research. In the sports club, the test site was separate from the children’s gross motor skills and the parents’ location, and test takers were divided into two groups: some were in charge of testing the children and some were in the presence of the parents to help them if they had any questions about the BRIEF questionnaire.

### 2.3. Instruments

A Behavior Rating Inventory of Executive Function (BRIEF) questionnaire and the Test of Gross Motor Development, second edition (TGMD-2) were used to collect data.

Behavior Rating Inventory of Executive Function (BRIEF) questionnaire: This test is a parent and teacher report, which is designed to measure the multidimensional nature of EFs based on behavior. In this study, we used the parent form (with 86 items) under 8 scales. In filling out the brief shapes, the parent or teacher will demonstrate whether the child has any issues with the pattern of behavior that a specific case is encountering. Behavior is evaluated as “never”, “sometimes”, or “often” with scores of “1”, “2”, or “3”, respectively. Hence, high scores on the BRIEF questionnaire indicate poor executive functioning [27]. Executive functions include the following subscales: emotional control (appropriately modulating emotional responses); inhibit (the ability to both control impulses and stop own behavior at the proper time); shift (free moving from one situation, activity, or aspect of a problem to another as the situation demands); plan/organize (the ability to anticipate future events, set goals, develop appropriate steps ahead of time, carry out tasks in a systematic manner, and understand and communicate the main ideas); working memory (simultaneously holding and processing short-term memory information); initiate (the ability to begin a task or activity and to generate ideas independently); organization of materials (being able to maintain relevant parts of the environment in a systematic manner); monitor (the ability to check work, assess performance, and keep track of your own and others’ efforts) [28]. These eight subscales form two composite indices: The Behavioral Regulation Index (BRI) and the Metacognition Index (MI) [29].

Test of Gross Motor Development, second edition (TGMD-2): The TGMD-2 is a qualitative measure used to assess the gross motor skills of children aged 3–10 years, and is one of the most common measurement tests in this area [15]. According to Burns et al. [30], content sampling, time sampling, and inter-scorer differences of the TGMD-2 were determined to be acceptable with coefficients of 0.87, 0.88, and 0.98, respectively [30,31]. Gross motor skills were assessed using the Test of Gross Motor Development, second edition (TGMD-2), the validity and reliability of which for young children has been established [31]. It is a process-based assessment, which means the qualitative aspects of skill performance are not measured quantitatively. Two subtests, known as locomotor and object control, formed the test. Each subtest is made up of six skills. The six locomotor skills are run, hop, gallop, leap, horizontal jump, and slide. The six object control skills are stationary dribble (bounce), striking a stationary ball, kick, underhand roll, catch, and overhand throw. In this test, participants execute each skill twice. Participants’ performance was video-recorded and all were analyzed by a prepared assessor agreeing on a score for each subtest, alluded to as locomotor and object control [32].

### 2.4. Data Analysis

Descriptive and inferential statistics were used for the statistical analysis. At first, the normality of the data using the Kolmogorov–Smirnov test and missing data were checked. Pearson correlation and multiple linear regression statistical methods were used to investigate the relationship between gross motor skills and executive function components. The level of significance was set at alpha <0.05. All statistical analyses were computed utilizing IBM SPSS Statistics for Windows, version 24.

## 3. Results

Before examining the hypotheses, the normality of data distribution was confirmed using Kolmogorov–Smirnov statistics. To examine the relationship between two subscales of gross motor skills as predictor variables, with eight criteria of executive functions (including inhibit, shift, emotion control, initiate, working memory, plan/organize, organization of materials, and monitor), a Pearson correlation coefficient (Table 1) and eight multiple linear regressions (Table 2) were separately used.

Considering that the correlation coefficients are located in a continuum [33], according to Table 1, it can be said that among the significant coefficients, there is a moderate correlation between object control and organization of materials, planning/organizing, working memory, and inhibition. There is also a moderate correlation between locomotor, organization of materials, and working memory and a weak correlation between locomotor and planning/organizing.

Next, we will separately discuss the multiple linear regression with the standard method for the ability to predict predictor variables (locomotor and object control) with the eight factors of executive functions as criterion variables.

Inhibit: The results of the multiple regression analysis of variance show that the model is significant and appropriate for this variable with values of F = 41.524 and *p* = 0.001. In addition, according to the set of R squared, it can be said that the success rate of this model is 47%, i.e., the selected model justifies 47% of the variance in the criterion variable (inhibit), so the model is a good model for examining gross motor skills.

Shift: The results of the multiple regression analysis of variance show that the model for this variable with values F = 0.129 and *p* = 0.879 is not significant and appropriate. Therefore, it can be said that gross motor skills do not have the ability to predict the shift factor.

Emotion control: The results of the multiple regression analysis of variance show that the model for this variable with values F = 0.730 and *p* = 0.485 is not significant and appropriate. Therefore, it can be said that gross motor skills do not have the ability to predict the emotion control factor.

Initiate: The results of the multiple regression analysis of variance show that the model for this variable with values F = 1.973 and *p* = 0.145 is not significant and appropriate. Therefore, it can be said that gross motor skills do not have the ability to predict the initiate factor.

Working memory: The results of the multiple regression analysis of variance show that the model is significant and appropriate for this variable with values of F = 30.539 and *p* = 0.001. In addition, according to the set of R squared, it can be said that the success rate of this model is 39%, i.e., the selected model justifies 39% of the variance in the criterion variable (working memory), and therefore the model is a good model for examining gross motor skills.

Plan/organize: The results of the multiple regression analysis of variance show that the model is significant and appropriate for this variable with values of F = 16.785 and *p* = 0.001. In addition, according to the set of R squared, it can be said that the success rate of this model is 25%, i.e., the selected model justifies 25% of the variance in the criterion variable (plan/organize), so the model is a good model for examining gross motor skills.

Organization of materials: The results of the multiple regression analysis of variance show that the model is significant and appropriate for this variable with values of F = 36.020 and *p* = 0.001. In addition, according to the set of R squared, it can be said that the success rate of this model is 43%, i.e., the selected model justifies 43% of the variance in the criterion variable (organization of materials), so the model is a good model for examining gross motor skills.

Monitor: The results of the multiple regression analysis of variance show that the model for this variable with values F = 3.295 and *p* = 0.042 is not significant and appropriate. Therefore, it can be said that gross motor skills do not have the ability to predict the monitor factor.

Significantly, 4 (inhibit, working memory, plan/organize, and organization factors, which had a success rate of 48%, 39%, 25%, and 43%, respectively) of the 8 models investigated the predictive power of gross motor growth factors. The other four models, which considered the four factors of executive functions (initiate, shift, emotion control, and monitor) as criterion variables, were not predicted under the influence of gross motor skills of the home environment and did not explain the appropriate model. Considering the significance of the four models (inhibit, working memory, plan/organize, and organization), in the following section we examine the predictive power of independent variables.

In detail, it can be stated that in the equation and model with the criterion of inhibition, out of the two predictor variables, the object control factor of gross motor skill factors with beta 0.655 has the ability to predict inhibition, meaning that changing a standard deviation in the factor variable object control causes the inhibitory variable to change by 0.655.

In the equation and model with the working memory criterion variable, out of the two predictor variables, the locomotor factor of gross motor skill factors with beta 0.559 has the ability to predict working memory, meaning that changing a standard deviation in the locomotor factor variable causes change in the working memory factor variable by 0.559.

Moreover, in the equation and model with the planning/organizing criterion variable, one of the predictor variables, the object control factor of the gross motor skill factors with beta 0.528 has the ability to predict planning/organizing, meaning that changing a standard deviation in the object control factor variable causes the programming factor variable to change by 0.528.

Finally, it can be said that in the equation and model with the organizational criterion variable, out of two predictor variables, the object control factor of gross motor skill factors with beta 0.601 has the ability to predict organization, meaning that changing a standard deviation in the object control factor variable causes the organization of materials factor variable to change by 0.601.

## 4. Discussion

The aim of this study was to investigate the relationship between executive functions and gross motor skills in rural children aged 8–10 years. A total of 93 elementary students were selected as subjects, and BRIEF and TGMD-2 tests were used. The results of this study show that there is a significant relationship between some factors of executive functions, such as inhibition, working memory, planning/organizing, organization of materials, and gross motor skills. However, no significant relationship was observed between other factors such as shift, emotion control, initiate and monitor, and gross motor skills.

The results of the correlations showed that there is a moderate correlation between object control and organization of materials, planning/organizing, working memory, and inhibition. There is also a moderate correlation between locomotor, organization of materials, and working memory and a weak correlation between locomotor and planning/organizing. In the following section, the results of the regression analysis will be discussed in detail. The results of this study on inhibition show that there is a positive and significant relationship between gross motor skills and inhibition. To be more precise, there is a positive and significant relationship among the components of gross motor skills; the object control skill in the Ulrich test includes stationary dribble (bounce), striking a stationary ball, kicking, underhand roll, catching, and overhand throw [34] with inhibition, which is one of the most important components of executive functions [35]. This means that children who had higher object control skills, on average, had a higher cognitive inhibition ability than their peers, which is important in relation to a completed task, when a goal is no longer appropriate, when an error needs to be corrected, as well as when appropriate stimuli are selected and inappropriate stimuli are rejected [36]. The results observed in this study are in line with the results of the Cook et al. [32] study conducted in the African low-income child community. The results from this study appear to suggest that gross motor skills are significantly and positively related to inhibition. They also stated that evidence determined from neuroimaging provides some clarification for this relationship, within the form of their co-activation amid task performance. Of course, it should be borne in mind that in the aforementioned research, inhibition was associated with both locomotor skills and object control skills. However, in this study, among the variables of gross motor skills only the object control agent was able to predict inhibition. The results also show that there is a positive and significant relationship between gross motor skills and working memory. More specifically, locomotor skills and working memory had a significant relationship, so that locomotor was able to predict working memory. It seems that working memory appeared to be related with locomotor skills, and perhaps related to the memory request of complex locomotor activities (such as those required in the TGMD-2; gallop, slide). Coordination demand requirements in locomotor skills, such as gallop and slide, are at a high level, i.e., involving concurrent body movements and movement sequences. As a result, it places a more prominent request upon the activation and sequencing of this data in working memory [37]. The results observed in the present study were completely consistent with the results of Cook et al. [32]. In their research, conducted in a low-income South African setting, they concluded that among the components of gross motor skills, locomotor skills (rather than object control) have the ability to predict working memory.

The results of research on planning/organizing and organization of materials showed that gross motor skills and organization of materials and planning/organizing as two components of executive functions have a positive and significant relationship. Specifically, object control skills, rather than locomotor skills, had the ability to predict the mentioned factors of executive functions. Piaget’s theory is the origin of the relation between gross motor skills and executive functions, and sensorimotor abilities as precursors for cognitive development [9]. Neuroimaging studies have given several interpretations for the links between motor and cognitive outcomes [18], showing that during cognitive and motor tasks, the cerebellum (critical for motor skills) and prefrontal cortex (critical for higher-order cognition) are co-activated [38]. However, due to the lack of research in the field of planning/organizing and organization of material factors, it seems that more research should be performed in this area.

Regarding the other components of executive functions, including shift, emotion control, and initiate and monitor, the results showed that there is no significant relationship between these components and gross motor skills. A possible explanation for the lack of significant relationship between these variables is that although many studies have acknowledged the relationship between cognition and movement, we can refer to the type of physical activity that the rural children participating in this study performed. Although executive function is not enhanced by unstructured physical activity, cognitively engaging physical activity that engages and challenges executive tasks, such as regular physical activity games and group sports, may be necessary to improve it [39]. Therefore, what is being done during the physical activity is more important than how much physical activity is happening.

Obviously, there are limitations of the study that should be mentioned. First, due to the fact that in this study a questionnaire was used to collect information about children’s executive functions, parental bias in answering questions is a limitation. Second, participants’ physical and mental well-being, as well as their motivation on the day of the test, may have affected their motor performance on the Ulrich test. It should also be noted that the present study was conducted with a small sample size and without a control group of urban children, which is another limitation. Finally, the researchers found a moderate to weak relationship, not a seemingly strong one.

### Research Suggestions

Given that the present study was conducted on both girls and boys, in order to control the effect of this factor, it is suggested that gender should be separately considered as an influential variable in future research. It is also suggested that research on the same subject be conducted using the urban children control group, albeit experimentally. Future studies should also address the effect of designing and implementing gross motor skills in order to make changes to children’s cognitive characteristics, especially executive functions. Additionally, different types of motor skills in specific classifications, such as muscular aspects (gross motor vs. fine motor skills), temporal aspects (discrete, serial, and continuous motor skills), environmental aspects (open vs. closed motor skills), and functional aspects (stability, manipulative, and locomotor tasks) could be used.

## 5. Conclusions

According to the results of this study, it seems that motor and cognitive development are related and, in many cases, motor development could predict the development of cognitive skills. Given the importance of cognitive development and executive functions in childhood, it seems that by helping to develop their gross motor skills, executive functions will also be strengthened. As we have seen in the present study, object control skills were able to predict inhibition, planning/organizing, and organization. On the other hand, locomotor skills predicted working memory. However, it should be noted that the relationship between the development of cognitive and motor skills is probably stronger and more significant in some aspects, while in others, motor skills are unpredictable. The reason for this seems to depend on factors such as the age of the study participants and their living environment, which subsequently affects the type of physical activity and other environmental factors. However, for a more accurate insight, more research is needed, especially on the components of executive functions that have received little research.

## Figures and Tables

**Table 1 healthcare-10-00616-t001:** Pearson correlation coefficient of two variables predicting gross motor skills with eight variables of performance criteria.

Variables	Object Control	Locomotor	Inhibit	Shift	Emotion Control	Initiate	Working Memory	Plan/ Organize	Organization of Materials	Monitor
Object control	1									
Locomotor	0.563 **	1								
Inhibit	0.691 **	0.432 **	1							
Shift	0.031	−0.019	0.290 **	1						
Emotion control	0.075	−0.042	0.328 **	0.689 **	1					
Initiate	0.082	−0.109	0.286 **	0.633 **	0.622 **	1				
Working memory	0.437 **	0.628 **	0.538 **	0.312 **	0.268 **	0.298 **	1			
Plan/ organize	0.521 **	0.285 **	0.597 **	0.457 **	0.478 **	0.488 **	0.544 **	1		
Organization of materials	0.661 **	0.445 **	0.618 **	0.212 *	0.264 *	0.298 **	0.577 **	0.571 **	1	
Monitor	0.147	−0.095	0.355 **	0.588 **	0.641 **	0.640 **	0.186	0.527 **	0.304 **	1

* *p* ≤ 0.05, ** *p* ≤ 0.01.

**Table 2 healthcare-10-00616-t002:** Prediction coefficients for the criterion variables.

Model		Unstandardized Coefficients		Standardized Coefficients	t	Sig.
		B	Std. Error	Beta		
1. Inhibit	(Constant)	−0.224	0.158		−1.416	0.160
	Object control	0.027	0.004	0.655	7.122	0.001 **
	Locomotor	0.004	0.005	0.064	0.692	0.491
2. Working memory	(Constant)	−0.456	0.154		−2.961	0.004 **
	Object control	0.005	0.004	0.123	1.245	0.216
	Locomotor	0.029	0.005	0.559	5.677	0.001 **
3. Plan/organize	(Constant)	0.157	0.178		0.882	0.380
	Object control	0.020	0.004	0.528	4.853	0.001 **
	Locomotor	−0.001	0.006	−0.013	−0.115	0.909
4. Organization of materials	(Constant)	−0.545	0.241		−2.262	0.026 *
	Object control	0.036	0.006	0.601	6.323	0.001 **
	Locomotor	0.009	0.008	0.107	1.122	0.265

* *p* ≤ 0.05, ** *p* ≤ 0.01.

## Data Availability

The data used to support the findings of current study are available from the corresponding author upon request.

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
