# Peer review of "The Relationship between Executive Functions and Gross Motor Skills in Rural Children Aged 8–10 Years"

_healthcare, 2022, doi:10.3390/healthcare10040616_

Round 1
Reviewer 1 Report
The Relationship between Executive Functions and Gross Motor Skills in Rural Children Aged 8-10 years
The paper sounds very interesting and relevant for the field, but several improvements are needed. Please, see my comments below:
Introduction
Based on recent studies, EFs are related to different brain regions (4).
The authors must include more references here to support that mentioned information.
However, research on the relationship between cognitive and motor development is limited to specific cognitive domains (9). (Line 54)
Please, include some examples to clarify this information.
In this regard and rejecting the relationship with the study of the effect of different dimensions of human development, especially cognitive and motor development, several studies have been conducted. (Lines 62 – 64). Please, include the references of those several studies.
It’s interesting how the authors mentioned “the rural children’s community” for the first time in the paper on page 77 and ahead in this part “Because of the environment, children from rural areas may have different opportunities to develop their skills (16). (Lines 78-79). But, very important information regarding the rationale behind these arguments is missing in the introduction. Why the environment of rural children’s community is different? Which different opportunities have they been compared to the urban children? The authors need to state those questions to show a clear conception for the readers of why is relevant to conduct such a study.
Also, it is concerning how the authors ended the introduction using the term “deprived rural children” (Line 84). Why are they deprived? Deprived of what? Please, clarify those questions and mention a proper contextualization about the concept of “deprived” you have adopted in your work.
Materials and Methods
Line 95: How the exclusion of behavioral, motor, or cognitive problems was made?
I would like to hear from the authors why they decided to apply a multiple linear regression using a small sample size of only 93 students without biasing the results? And please, switch the term ‘student’ for ‘schoolchild’ over the paper. It is more adequate in this age range.
I found no information regarding sample size calculation.
Results and Discussion
I’m very concerned about the choices made by the authors to discuss their results. For example, the results of Cook et al. (2019) are used to compare to the current study, but the study conducted in South Africa used a sample of preschool children from urban and rural low-income settings, which was not used in the current student.
The authors used Pearson correlation for data analysis, but no information on correlation coefficient power is shown in the paper. A positive and significant correlation means nothing without a proper strength description. How strong or weak were the values of the correlations found in the study?
Finally, despite the authors highlighting some limitations at the end of the discussion I must suggest more: (i) The small sample size; (ii) The absence of a control group of children from the urban areas; (iii) No presence of strong correlations, only moderate and weak.
Conclusion
The authors pointed out various arguments that were not investigated or discussed in the manuscript. Those are very speculative and need improvements.
Author Response
Dear Reviewer,
I attach the answers.
Thank you!

Reviewer 2 Report
Comments to the authors:
Thank you for the opportunity to read this work. It is a very interesting topic of study.
However, some areas of this paper would need enhancing.
Abstract:
Authors should consider adding a sentence that substantiates the purpose of the study (before the study aim): Why is it relevant to study the relationship between executive functions and gross motor skills in rural children aged 8-10 years?
Introduction:
The same question - Why is it relevant to study the relationship between executive functions and gross motor skills in rural children aged 8-10 years? – is not well answered in the study introduction. Besides the scarcity of studies in this domain, understanding this specific relationship in rural children has to add something to practice or subsequent studies.
In line 67 authors had referred to studies conducted by Adele Diamond. Is not a he is a she. I suggest some care with how you use pronouns in scientific writing. It may be important to review this throughout the article to ensure accuracy beyond what might be considered style.
Methods:
Line 92 and 93: Is not clear what is a letter of satisfaction. It is a consent letter? Ethical considerations in the “Subjects and Design” it is lacking.
Line 99 to 102: Is not clear how the procedure was run. Please provide a more complete description. It is important to ensure that this study may be replicated by another research team.
Line 134 and 135: The sentence seems to have been interrupted in the middle and rewritten. I do not be able to understand this specific part of the test.
Overall, the description of the measures should be revised to a more comprehensive way.
Results:
I recommend that the results be reported according to APA rules.
Discussion:
In the discussion, it was not clear the relevance of study results.
Again, why is important to study this relationship in rural children? The improvement in gross motor skills and executive functions will have some level of impact on children's health, development, school achievement?
Research suggestion is lacking more substance, more reflection.
Some major spelling/grammar errors.
The manuscript requires a careful review.
Author Response

(The authors gave the same response as above.)

Round 2
Reviewer 1 Report
Dear authors,
I am really impressed with all improvements made throughout the paper. However, before considering this manuscript for publication I still have some concerns described below:
1 - Please, include all the information regarding sample size calculation in the paper. The authors are still hiding important information, such as power, confidence interval, estimated effect, etc.
2 - Regarding this part:
"Considering the living environment, rural children have a wider space to play than urban children do. Usually they live in villas, while urban children live in apartment houses without yards and park's playground equipment"
Please, add some more considerations taking into account the term "opportunity to explore executive functions and motor skills". Do you think rural children have more both than urban children?
3 - Finally, explore more details regarding moderate and weak correlations effects into the discussion.
Looking forward to hearing from the authors soon.
Reviewer 2 Report
Thank you for addressing the comments from the previous feedback round. I am largely satisfied with your responses and amendments to the paper.
However, there still seems to me to be some inconsistency in the purpose of the study.
Between lines 81 and 101 the authors some sentences are confusing and lack clarity, namely:
"Some studies have confirmed that parents' socioeconomic status and the child's living environment may be related to children's motor development [21]. According to these studies-83, it is true that motor skills affect the child's subsequent development, as they predict 84 other aspects of development. However, children's motor development is also affected by 85 various factors, including social and economic status [22], and in the meantime, the rural community of 86 children must be considered. "
Regarding the variable children from rural backgrounds, within the same line range, it remains unclear whether the authors consider it relevant to study this topic because this context is a protective factor ("Because of the environment, children from rural areas may have different opportunities to develop their skills [21]. Considering the living environment, rural children have a wider space to play than urban children do. Usually, they live in villas, while urban children live in apartment houses without yards and park's playground equipment.") or risk ("However, children's motor development is also affected by 85 various factors, including social and economic status [22], and in the meantime, the rural children's community should be considered.").
I warn, finally, for the authors to be careful how these arguments are described to avoid the suggestion of prejudice.
